# Controllable Text Generation with Neurally-Decomposed Oracle

**Tao Meng**[*]
University of California, Los Angeles
tmeng@cs.ucla.edu

**Sidi Lu**[*]
University of California, Los Angeles
sidilu@cs.ucla.edu

**Nanyun Peng**
University of California, Los Angeles
violetpeng@cs.ucla.edu

**Kai-Wei Chang**
University of California, Los Angeles
kwchang@cs.ucla.edu

## Abstract

We propose a general and efficient framework to control auto-regressive generation models with NeurAlly-Decomposed Oracle (NADO). Given a pre-trained base language model and a sequence-level boolean oracle function, we propose to decompose the oracle function into token-level guidance to steer the base model in text generation. Specifically, the token-level guidance is approximated by a neural model trained with examples sampled from the base model, demanding no additional auxiliary labeled data. Based on posterior regularization, we present the closed-form optimal solution to incorporate the token-level guidance into the base model for controllable generation. We further provide a theoretical analysis of how the approximation quality of NADO affects the controllable generation results. Experiments conducted on two tasks: (1) text generation with lexical constraints and (2) machine translation with formality control demonstrate that our framework efficiently guides the base model towards the given control factors while maintaining high generation quality.

## 1 Introduction

Auto-regressive language models have been widely used for text generation. With the recent development of large-scale pre-trained language models (Radford et al., 2019; Brown et al., 2020; Raffel et al., 2020; Lewis et al., 2020), they have achieved state-of-the-art performances in applications such as machine translation (Bahdanau et al., 2015; Luong et al., 2015), image captioning (Anderson et al., 2018; You et al., 2016) and open domain text generation (Zhang and Lapata, 2014; Yao et al., 2019; Vinyals and Le, 2015; Shang et al., 2015; Lu et al., 2018). However, many applications such as open-domain creative generation (Yao et al., 2019; Goldfarb-Tarrant et al., 2020; Tian and Peng, 2022; Han et al., 2022; Chen et al., 2022; Spangher et al., 2022) require to control model output with specific sequence-level attributes. The attributes can be specified by a set of rules[2] or by an abstract concept (e.g., the generated text follows a particular writing style). How to control auto-regressive language models to satisfy these attributes is an open challenge.

In this paper, we propose a general and flexible framework for controllable text generation. Given a base pre-trained language model and a sequence-level oracle function indicating whether an attribute is satisfied, our goal is to guide the text generation to satisfy certain attributes using the oracle. To

---

[*]equal contribution

[2]For example, lexical constraints require certain words to appear in the generated text (Hokamp and Liu, 2017; Lin et al., 2020)

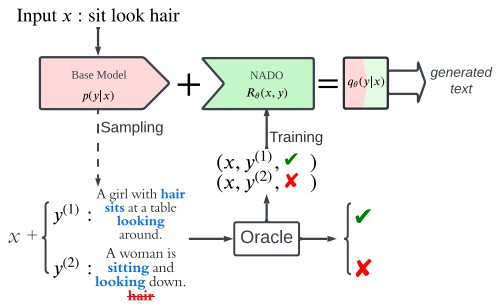

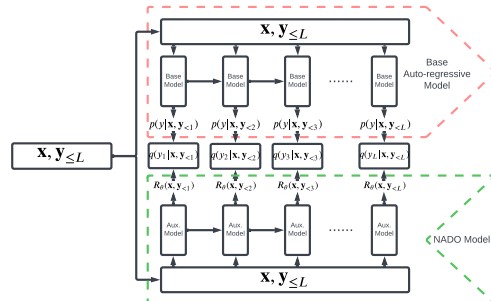

(a) Take lexically constrained generation as an example, where the oracle checks whether all keywords in the input $\mathbf{x}$ are incorporated in generated text $\mathbf{y}$. With proper training using samples from the base model $p$ (dashed arrow) labeled by the oracle, we decompose the oracle into token-level guidance and parameterize it by an auxiliary model $R_\theta$ (NADO). We use $R_\theta$ to provide guidance when generating text with the base model (see details in Fig. 1(b)).

(b) Illustration of the controlled generation process. Both the base model and the auxiliary model (NADO) take input $\mathbf{x}$ and the generated sequence (prefix) $\mathbf{y}_{<L}$ as input. The base model, in each step, outputs a token distribution $p(y_i|\mathbf{x}, \mathbf{y}_{<i})$. Guided by NADO $R_\theta$, we obtain the distribution $q$ (See Sec. 3.2), based on which we generate the output token.

Figure 1: Illustration of pipeline incorporating NADO (left) and model architecture (right).

this end, we propose to decompose the sequence-level oracle into token-level guidance, such that when generating the $i-$th token in the output sequence given the prefix, instead of sampling from the base model, we modify the probability distribution of the output token based on the token-level guidance. Specifically, we formulate the control as an optimization problem based on posterior regularization (Ganchev et al., 2010) and solve the closed-form optimal solution to incorporate the token-level guidance for text generation. The decomposition is approximated by an auxiliary neural network model, called NeurAlly-Decomposed Oracle (NADO), which is trained on data sampled from the base model and supervised by the sequence-level oracle (see the illustration Fig. 1a). We further provide theoretical analysis on how NADO's approximation quality affects the controllable generation results. Note that in the entire process, we treat the base model and the sequence-level oracle as black-box functions, without the need for any refactoring or fine-tuning.

A few existing controllable generation works (e.g., Lu et al. (2021, 2022b)) design search algorithms for generating texts with *lexical constraints*. However, their approaches cannot generally be applied to constraints such as style. Another line of work such as PPLM (Dathathri et al., 2020), GeDI (Krause et al., 2021), and FUDGE (Yang and Klein, 2021) also aim to guide the base model with an auxiliary model. However, they either shift the base model distribution in a post-hoc manner without theoretical guarantee, or/and require external labeled data to train the auxiliary model. Khalifa et al. (2021); Korbak et al. (2022) propose a generation with distributional control approach. Our control objective derived through posterior regularization resembles their energy-based model representation. However, they approximate the energy-based model using a KL-adaptive distributional policy, while we propose to decompose the sequence-level oracles into token-level approximated by NADO. With the decomposition, base models receive explicit controlling signal in generating every token from the oracle. Furthermore, since NADO is trained on the data sampled from the base models, it aligns better with the base model's distribution and thus can achieve better control.

We conduct experiments on lexically constrained generation (LCG) tasks and a machine translation (MT) formality change task. In LCG tasks, the oracle is a rule-based keyword checker. We achieve almost perfect keyword incorporation with significantly boosted BLEU scores compared to previous approaches that design specific decoding algorithms (Lu et al., 2021). In the formality-controlled MT task, we are provided with a formality oracle predicting whether a sentence is formal or not, and the goal is to guide the model to generate formal translations. Compared with recent work (Yang and Klein, 2021), we improve the BLEU score by 3 points as well as improve the formality rate, demonstrating NADO's superior ability to incorporate external oracle supervision. Both experiments demonstrate the effectiveness of our framework in dealing with various types of control while maintaining high-quality generation results.[3]

---

[3]Our code can be found at `https://github.com/MtSomeThree/constrDecoding`.

## 2 Related Work

**Controllable Text Generation with Auto-regressive Models.** Most previous work on controllable text generation are based on the auto-regressive framework. Zhang et al. (2022) summarize these methods into three categories: fine-tuning, refactor/retraining and post-processing. The first two categories, e.g., fine-tuning with control code (Peng et al., 2018; Keskar et al., 2019) or prompt-based methods (Sheng et al., 2020; Shin et al., 2020; Lester et al., 2021; Li and Liang, 2021), are usually weaker in controllability and inefficient in training considering the size of language models are dramatically increasing nowadays. Generally, the post-processing methods are considered expensive in inference and low quality in generated texts. However, our framework, as a kind of post-processing method, is able to achieve high generation quality demonstrated in the experiments and maintains efficient inference.

**Controllable Text Generation via Post-processing.** There are two major lines in post-processing: (1) modifying the decoding algorithm and (2) guiding generation with an auxiliary model. For some token-level controlled generation tasks like lexically constrained generation, we can inject the constraints into the decoding algorithm (e.g., constrained beam search (Anderson et al., 2017; Post and Vilar, 2018) and NeuroLogic decoding (Lu et al., 2021, 2022b)). Though shown effectiveness in lexically constrained generation, these algorithmic methods fail to fundamentally touch the token distribution, and are hard to handle other abstract attributes.

In the second line, PPLM (Dathathri et al., 2020) proposes an auxiliary discriminator for the expected attribute to guide the model; GeDi (Krause et al., 2021) and DEXPERTS (Liu et al., 2021) apply contrastive learning and train an auxiliary language model to reweight the token distribution in each step; Plug-and-Blend (Lin and Riedl, 2021) further extends the GeDi framework by adding a planner architecture. FUDGE (Yang and Klein, 2021) leverages external token-level oracle to train a discriminator for guiding the base model. These methods either require external token-level oracle guidance or auxiliary labeled datasets to train the auxiliary models. However, the distribution of the data used to train the auxiliary model is different from what the based model is trained on. This distributional discrepancy causes the drop of generating quality as we will show in the experiments. For example, given a controlling attribute $a$, Fudge generates next token $y_i$ based on Bayesian rule $P(y_i|\mathbf{y}_{<i}, a) \propto P(a|\mathbf{y}_{\leq i})P(y_i|\mathbf{y}_{<i})$. However, their $P(a|\mathbf{y}_{\leq i})$ and $P(y_i|\mathbf{y}_{<i})$ are not estimated based on the same distribution. In contrast, NADO is trained with data sampled from the base model. Therefore, it learns to incorporate with the base model, which avoids the distributional discrepancy. We also provide a principle, theoretical framework to discuss the optimal solution of incorporating the sequence-level oracle.

## 3 Methodology

We approach the sequence-level controllable text generation problem by decomposing the sentence-level oracle into token-level guidance. We formulate this as an optimization problem. Since the token-level guidance is intractable, we propose to train an auxiliary model, called NeurAlly-Decomposed Oracle (NADO), to approximate it. During the inference time, NADO guides the base model to generate sequences that satisfy the oracle constraints.

In the rest of this section, we discuss 1) the formulation to decompose the sequence-level oracle function into token-level guidance; 2) the formulation to incorporate the token-level guidance into the base model to achieve control; 3) the approximation of the token-level guidance using NADO; 4) a theoretical analysis of the impact of NADO approximation to the controllable generation results; and 5) the training of NADO.

### 3.1 Setup: Notations and Problem Formulation

We use $\mathbf{x} \in \mathcal{X}$ to denote the input and $\mathbf{y} \in \mathcal{Y}$ to denote the generated sequence. $y_i$ is the $i-$th token in $\mathbf{y}$ and $\mathbf{y}_{<i}$ is the sequence prefix from the beginning to the $(i-1)-$th token. We denote the base auto-regressive generation model as $p(y_i|\mathbf{x}, \mathbf{y}_{<i})$, hence the sequence-level distribution is given by $p(\mathbf{y}|\mathbf{x}) = \prod_i p(y_i|\mathbf{x}, \mathbf{y}_{<i})$. A sequence-level oracle is defined as a boolean function $C : \mathcal{X} \times \mathcal{Y} \to \{0, 1\}$. We formalize the optimization objective based on posterior regularization (Ganchev et al., 2010). Basically, we explore a token-level distribution $q^*(y_i|\mathbf{x}, \mathbf{y}_{<i})$ and its corresponding sequence-level distribution $q^*(\mathbf{y}|\mathbf{x})$, satisfying

1. $q^*(\mathbf{y}|\mathbf{x}) = \prod_i q^*(y_i|\mathbf{x}, \mathbf{y}_{<i})$, i.e., $q^*$ can be treated as an auto-regressive model.
2. $q^*(\mathbf{y}|\mathbf{x}) = 0$ if $C(\mathbf{x}, \mathbf{y}) = 0$, i.e., $q^*$ only generates sequences satisfying the oracle $C$.
3. Given an input $\mathbf{x}$, $KL(p(\mathbf{y}|\mathbf{x})\|q^*(\mathbf{y}|\mathbf{x}))$ is minimized, i.e., $q^*$ should be as similar to the base model as possible.

Khalifa et al. (2021); Korbak et al. (2022) derive a similar optimization formulation as property 2, 3, to represent constraints through energy-based models and approximate it with distributional policy gradient. In this work, we propose to decompose oracle to token-level guidance to steer the generation. We discuss our approach in the following.

### 3.2   Token-level Guidance and Closed-Form Solution For $q^*$

Before we compute the solution for $q^*$, given the base model $p$ and oracle $C$, we first define the token-level guidance as a success rate prediction function $R_p^C(\mathbf{x})$, which defines the probability of the sequence generated by $p$ satisfies the oracle $C$ given the input $\mathbf{x}$. We similarly define $R_p^C(\mathbf{x}, \mathbf{y}_{\leq i})$ as the probability of success given input $\mathbf{x}$ and prefix $\mathbf{y}_{<i}$. By definition, we have

$$R_p^C(\mathbf{x}) = \Pr_{\mathbf{y} \sim p(\mathbf{y}|\mathbf{x})}[C(\mathbf{x}, \mathbf{y}) = 1] = \sum_{\mathbf{y} \in \mathcal{Y}} p(\mathbf{y}|\mathbf{x})C(\mathbf{x}, \mathbf{y})$$

$$R_p^C(\mathbf{x}, \mathbf{y}_{\leq i}) = \Pr_{\mathbf{y} \sim p(\mathbf{y}|\mathbf{x})}[C(\mathbf{x}, \mathbf{y}) = 1|\mathbf{y}_{<i}] = \sum_{\mathbf{y} \in \mathcal{Y}} p(\mathbf{y}|\mathbf{x}, \mathbf{y}_{<i})C(\mathbf{x}, \mathbf{y}). \tag{1}$$

With the function $R_p^C$, we now derive the closed-form solution of $q^*$ considering conditions 2 and 3 defined in Sec. 3.1. Given input $\mathbf{x}$, we define the feasible sequence-level distribution set $Q$ as

$$Q := \{q| \sum_{\mathbf{y}:\ C(\mathbf{x},\mathbf{y})=0} q(\mathbf{y}|\mathbf{x}) = 0\}, \tag{2}$$

then the sequence-level closed-form solution for $q^*$ is given by

$$q^*(\mathbf{y}|\mathbf{x}) = \arg\min_{q \in Q} KL(p(\mathbf{y}|\mathbf{x})\|q(\mathbf{y}|\mathbf{x})) = \frac{p(\mathbf{y}|\mathbf{x})C(\mathbf{x}, \mathbf{y})}{R_p^C(\mathbf{x})}. \tag{3}$$

Considering condition 1 in Sec. 3.1 to make $q^*$ tractable, we decompose $q^*(\mathbf{y}|\mathbf{x})$ into token-level. The closed-form solution is given by

$$q^*(y_i|\mathbf{x}, \mathbf{y}_{<i}) = \frac{R_p^C(\mathbf{x}, \mathbf{y}_{\leq i})}{R_p^C(\mathbf{x}, \mathbf{y}_{\leq i-1})} p(y_i|\mathbf{x}, \mathbf{y}_{<i}). \tag{4}$$

The decomposition is unique. The proof and detailed derivation can be found in the appendix.

**Control with Soft Constraints.** In Eq. (2) we define the feasible distribution set as distribution that the possibility of a sequence violate the oracle function is $0$. However, in some applications, we expect to control the generation with soft constraints. For example, we want the model to generate sentence about sports with probability $r = 0.8$. Our framework also supports controlling the generation with soft constraints. To achieve this, with a pre-defined ratio $r \in [0, 1]$, we alternatively define a general feasible set $Q$ as

$$Q := \{q| \sum_{\mathbf{y}:\ C(\mathbf{x},\mathbf{y})=1} q(\mathbf{y}|\mathbf{x}) = r\},$$

where Eq. (2) is the special case when $r = 1$. The general token-level closed-form solution is

$$q^*(y_i|\mathbf{x}, \mathbf{y}_{<i}) = \frac{\alpha R_p^C(\mathbf{x}, \mathbf{y}_{\leq i}) + \beta(1 - R_p^C(\mathbf{x}, \mathbf{y}_{\leq i}))}{\alpha R_p^C(\mathbf{x}, \mathbf{y}_{\leq i-1}) + \beta(1 - R_p^C(\mathbf{x}, \mathbf{y}_{\leq i-1}))} p(y_i|\mathbf{x}, \mathbf{y}_{<i}),$$

where $\alpha = \frac{r}{R_p^C(\mathbf{x})}, \beta = \frac{1-r}{1-R_p^C(\mathbf{x})}$.

Similar to Eq. (4), once we have access to $R_p^C$, we can directly compute the closed-form solution even though the form is much more complicated. In this paper we only focus on hard constraints ($r = 1$), however, here we demonstrate that our framework is capable of handling soft constraints as well.

### 3.3 Approximating $R_p^C$ by NADO and Theoretical Analysis

Unfortunately, function $R_p^C$ defined in Eq. (1) is intractable. We cannot enumerate all possible sequences $\mathbf{y}$ since the space is exponentially large and essentially infinite. Hence, we train a neural model NADO to approximate this well-defined function. We use $R_\theta^C$ to denote NADO parameterized by $\theta$. In this section, we derive bounds to provide a theoretical analysis about the correlation between errors in approximation and errors in corresponding sequence-level distribution. Generally, when $R_\theta^C$ approximates $R_p^C$ precisely enough, we have an upper bound for the sequence-level distribution discrepancy. The following lemma provides the formal definition.

**Lemma 1** We define distribution

$$q(y_i|\mathbf{x}, \mathbf{y}_{<i}) \propto \frac{R_\theta^C(\mathbf{x}, \mathbf{y}_{\leq i})}{R_\theta^C(\mathbf{x}, \mathbf{y}_{\leq i-1})} p(y_i|\mathbf{x}, \mathbf{y}_{<i}). \tag{5}$$

If there exists $\delta > 1$ such that given input $\mathbf{x}$, $\forall \mathbf{y}_{<i}$, $\frac{1}{\delta} < \frac{R_\theta^C(\mathbf{x}, \mathbf{y}_{\leq i})}{R_p^C(\mathbf{x}, \mathbf{y}_{\leq i})} < \delta$, we have

$$KL(q^*(\mathbf{y}|\mathbf{x})\|q(\mathbf{y}|\mathbf{x})) < (2L + 2)\ln\delta,$$

where $L$ is the length of the sequence $\mathbf{y}$.

We also notice that by definition, $R_p^C$ satisfies the following equation:

$$\sum\nolimits_{y_i} R_p^C(\mathbf{x}, \mathbf{y}_{\leq i})p(y_i|\mathbf{x}, \mathbf{y}_{<i}) = R_p^C(\mathbf{x}, \mathbf{y}_{\leq i-1}). \tag{6}$$

If $R$ also satisfies Eq. (6), we can tighten this bound. Formally,

**Lemma 2** Given the condition in Lemma 1, if $q$ is naturally a valid distribution without normalization (i.e., $\sum_{y_i} \frac{R_\theta^C(\mathbf{x}, \mathbf{y}_{\leq i})}{R_\theta^C(\mathbf{x}, \mathbf{y}_{\leq i-1})} p(y_i|\mathbf{x}, \mathbf{y}_{<i}) = 1$), we have

$$\forall x, KL(q^*(\mathbf{y}|\mathbf{x})\|q(\mathbf{y}|\mathbf{x})) < 2\ln\delta.$$

This lemma shows that with the auto-regressive property, the error does not accumulate along with the sequence. The proof is in the appendix. These two bounds indicate that when training the model $R_\theta^C$, we should push it to satisfy Eq. (6) while approximating $R_p^C$.

### 3.4 Training NADO

In Fig. 1b we show the architecture of NADO. In general, NADO can be any seq2seq model. During training, it takes $\mathbf{x}, \mathbf{y}$ as input and predicts from $R_\theta^C(\mathbf{x}, \mathbf{y}_{\leq 0})$ to $R_\theta^C(\mathbf{x}, \mathbf{y}_{\leq T})$. During the inference time, there are two parallel forward pass[4] to compute the token distribution $q$. Considering the size of the NADO is usually much smaller than the base model, the whole forward pass takes no more than 2x base model forward pass time.

Now we discuss the training objective. In training, with some predefined input distribution $\mathcal{X}$, we sample $\mathbf{x} \sim \mathcal{X}$, $\mathbf{y} \sim p(\mathbf{y}|\mathbf{x})$. We take these sampled $(\mathbf{x}, \mathbf{y})$ pairs as training examples, and use the boolean value $C(\mathbf{x}, \mathbf{y})$ as their labels for all steps. We use cross entropy (denoted as $CE(\cdot, \cdot)$) as the loss function, formally, $L_{CE}(\mathbf{x}, \mathbf{y}, R_\theta^C) = \sum_{i=0}^{T} CE(R_\theta^C(\mathbf{x}, \mathbf{y}_{\leq i}), C(\mathbf{x}, \mathbf{y}))$. Given a particular input $\mathbf{x}$, in expectation, we have

$$\begin{aligned}
\mathbb{E}_{\mathbf{y} \sim p(\mathbf{y}|\mathbf{x})} L_{CE}(\mathbf{x}, \mathbf{y}, R_\theta^C) &= \sum\nolimits_{\mathbf{y} \in \mathcal{Y}} p(\mathbf{y}|\mathbf{x}) L_{CE}(\mathbf{x}, \mathbf{y}, R_\theta^C) \\
&= \sum_{i=0}^{T} R_p^C(\mathbf{x}, \mathbf{y}_{\leq i})\log R_\theta^C(\mathbf{x}, \mathbf{y}_{\leq i}) + (1 - R_\theta^C(\mathbf{x}, \mathbf{y}_{\leq i}))\log(1 - R_\theta^C(\mathbf{x}, \mathbf{y}_{\leq i})) \\
&= \sum_{i=0}^{T} CE(R_p^C(\mathbf{x}, \mathbf{y}_{\leq i}), R_\theta^C(\mathbf{x}, \mathbf{y}_{\leq i}))
\end{aligned}$$

$$(7)$$

---

[4]In practice, to avoid enumerating the vocabulary, $R_\theta^C$ outputs a vector over vocabulary (i.e., $R_\theta^C(\mathbf{x}, \mathbf{y}_{\leq i-1} \oplus y)$ for all possible $y$, $\oplus$ is the concatenation operation), then we can directly do element-wise multiplication between $R_\theta^C$ and $p$.

Therefore, $L_{CE}$ empirically estimates the cross entropy loss between $R_\theta^C$ and the ground truth $R_p^C$ which is intractable.

As we analyze above, we also regularize $R_\theta^C$ for satisfying Eq. (6) based on KL-divergence:

$$L_{reg}(\mathbf{x}, \mathbf{y}, R_\theta^C) = f_{KL}\left(\sum_{y_i} R_\theta^C(\mathbf{x}, \mathbf{y}_{\leq i})p(y_i|\mathbf{x}, \mathbf{y}_{<i}), R_\theta^C(\mathbf{x}, \mathbf{y}_{\leq i-1})\right).$$

$f_{KL}(p, q) = p\log\frac{p}{q} + (1-p)\log\frac{1-p}{1-q}$ is KL-divergence regarding $p$ and $q$ as two Bernoulli distributions. We use a hyper-parameter $\lambda > 0$ to balance these losses. The final training loss is

$$L(\mathbf{x}, \mathbf{y}, R_\theta^C) = L_{CE}(\mathbf{x}, \mathbf{y}, R_\theta^C) + \lambda L_{reg}(\mathbf{x}, \mathbf{y}, R_\theta^C). \tag{8}$$

### 3.5 Sampling

In Sec. 3.4 we describe that we train NADO by sampled data from base model $p$. One advantage is that we are able to leverage different sampling strategies to better adapt to different application scenarios. It is also possible to leverage reinforcement learning to train $R_\theta^C$, and we discuss our connection to reinforcement learning in the appendix. In this section, we introduce two sampling strategies and their corresponding properties.

**Sampling with Temperature Control.** In some task, the output sequences are not diverse much, in other words, the token distribution in each step is very peaky. Since our NADO is trained on the sampled examples, we expect those examples to cover as much tokens combination as possible to avoid overfitting. Therefore, we add temperature factor $T$ to smooth the distribution (Ackley et al., 1985). Specifically, we sample $\mathbf{y}$ from distribution $p(\mathbf{y}|\mathbf{x})^{\frac{1}{T}}$, and add coefficient $p(\mathbf{y}|\mathbf{x})^{1-\frac{1}{T}}$ when computing the cross-entropy loss. Formally, the expected loss is

$$\mathbb{E}_{\mathbf{y}\sim p(\mathbf{y}|\mathbf{x})^{\frac{1}{T}}}\left[p(\mathbf{y}|\mathbf{x})^{1-\frac{1}{T}}L_{CE}(\mathbf{x}, \mathbf{y}, R_\theta^C)\right] = \sum_{\mathbf{y}\in\mathcal{Y}} p(\mathbf{y}|\mathbf{x})L_{CE}(\mathbf{x}, \mathbf{y}, R_\theta^C),$$

which is same as the original expected loss in Eq. (7).

**Importance Sampling.** In practice, the training process of NADO can be extraordinarily difficult when samples generated by the base model $p$ hardly satisfy $C$. i.e. $\mathbb{E}_{\mathbf{y}\sim p(\mathbf{y}|\mathbf{x})}[p(C|\mathbf{x}, \mathbf{y})] \simeq 0$. Hence, we introduce the importance sampling (Hammersley and Morton, 1954) to tackle this issue. Basically, we leverage existing partially trained $\hat{R}_\theta$ to form distribution $\hat{q}$. Although $\hat{R}_\theta$ is not well-trained, it is still able to provide positive guidance to produce samples satisfying $C$. Note that $\hat{q}$ does not have to be updated in each training epoch. With coefficient $\frac{p(\mathbf{y}|\mathbf{x})}{\hat{q}(\mathbf{y}|\mathbf{x})}$, the expected loss is same as the original expected loss:

$$\mathbb{E}_{\mathbf{y}\sim\hat{q}(\mathbf{y}|\mathbf{x})}\left[\frac{p(\mathbf{y}|\mathbf{x})}{\hat{q}(\mathbf{y}|\mathbf{x})}L_{CE}(\mathbf{x}, \mathbf{y}, R_\theta^C)\right] = \sum_{\mathbf{y}\in\mathcal{Y}} p(\mathbf{y}|\mathbf{x})L_{CE}(\mathbf{x}, \mathbf{y}, R_\theta^C).$$

## 4 Experiments

We conduct experiments on two tasks: lexically constrained generation (LCG) and machine translation (MT) with formality change. For the former, we use GPT-2 (Radford et al., 2019) as the base model and for the latter, we use a sequence-to-sequence model, MarianMT (Junczys-Dowmunt et al., 2018). We demonstrate our framework is generally effective in both scenarios. The boolean oracle is a rule-based function checking whether all lexical constraints are satisfied in LCG task, while in MT it is a classifier trained on an external dataset identifying the formality of the text. We put all details about hyper-parameter settings in the appendix.

### 4.1 Text Generation with Lexical Constraints

We evaluate our model on two general classes of LCG problems:

- Unsupervised LCG: annotation for lexical constraints are not available during training, but are expected to be in their exact order and lexical form during inference.

- Supervised LCG: annotation for lexical constraints are available, yet the words may appear in a different lexical form (e.g., "look" can appear in the past tense "looked") or a different order in the generated text.

In both cases, we define oracle $C$ as a boolean function indicating whether the generated sequence satisfies all of the lexical constraints. We do not naturally have negative samples (i.e. the sequences that do not satisfy all constraints) to train the auxiliary model in both settings, thus, it is non-trivial to compare against methods requiring both positive and negative labeled data for training the auxiliary model like FUDGE and GeDi.

**Data Setup** For unsupervised LCG, we follow the settings in POINTER (Zhang et al., 2020) and conduct our experiments on Yelp! Review and News dataset. Each of the unsupervised LCG dataset contains a great number of un-annotated, raw sequences for training (160K for Yelp! Review and 268,586 for News). During inference, the model is expected to generate text lexically constrained in the exact order and form by a specific number of keywords (7 for Yelp! Review and 4 for News). For supervised LCG, we evaluate the proposed method on CommonGen (Lin et al., 2020). CommonGen is a supervised LCG task that aims to examine the commonsense of neural text generation models. For training, it contains 32,651 unique key concepts (i.e. the constraints) with 67,389 completed sequences in total. It also contains a validation set with 993 concepts and 4018 reference sequences. For a more robust evaluation, the dataset maintains an open leaderboard that benchmarks different approaches on a withheld test set. We follow most of the data configurations specified in the original paper that first introduced the datasets.

**General Model Setup** We investigate the effectiveness of different factors in our framework by enumerating different combinations of them. We implement two types of base model:

- (Seq2seq base model) A sequence-to-sequence model $p(\mathbf{y}|\mathbf{x})$ that takes into account the lexical constraints as condition sequence input;
- (DA base model) A language model that is only domain-adapted to $p(\mathbf{y})$ but unconditioned on anything. This is a challenging setting, since we impose the lexical constraints only with NADO. This setting is to better verify the effectiveness and efficiency of the proposed method and control irrelevant factors.

Under both $p(\mathbf{y}|\mathbf{x})$ and $p(\mathbf{y})$ settings, we fine-tune the base model from the pre-trained GPT2-Large.

During training, NADO is trained as a Seq2seq-like model[5], which takes in the keys (for unsupervised LCGs, they are generated by randomly sampling a specific number of natural words in the original sentence) and generates the token-level guidance $R_\theta^C(\mathbf{x}, \mathbf{y}_{\leq i})$. For each pseudo key, we sample 32 target text with top-p ($p = 0.8$) random sampling from base model $p$. We conduct experiments to test different training setups for NADO:

- (NADO training) The proposed training process described in Sec. 3.4.
- (Warmup) We warm up NADO by maximizing the likelihood of positive samples, but only backpropagating the gradient to the parameters of $R_\theta$. The warm-up $R_\theta^C$ is used for importance sampling described in Sec. 3.5. With DA base models, however, the warmup process is always incorporated for practical success of training (see the results for DA pretrained w/o warmup).

We also consider the setting with warmup only, which can be treated as a stronger baseline to verify that the major improvement of our framework is not coming from the extended capacity in NADO.

**Results and Analysis** We compare the performance under different setups of our model to previous state-of-the-art methods on LCG tasks, including insertion-based models (Levenshtein Transformer (Gu et al., 2019) with Lexical Constraints (Susanto et al., 2020), InsNet (Lu et al., 2022a), etc.) and decoding-based algorithms. We also compare the results with a simple baseline which address the problems with a standard Seq2seq pipeline. The results are as shown in Table 1.

NADO consistently improves the BLEU score and coverage in different setups. Furthermore, under the best setting of each task (see bolded items in the table), NADO performs significantly better than most baselines in generation quality and can achieve very good lexical constraints coverage rate.

---

[5]In this experiment, the input $\mathbf{x}$ is only describing the lexical constraint $C$. However, our framework also supports general inputs in other Seq2seq tasks with constraints. For example, machine translation with lexical constraints where the constraint $C$ is different from the input $\mathbf{x}$.

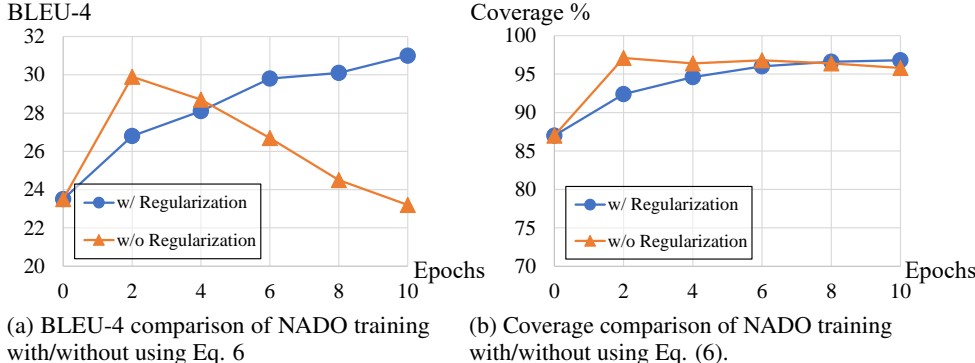

(a) BLEU-4 comparison of NADO training with/without using Eq. 6

(b) Coverage comparison of NADO training with/without using Eq. (6).

Figure 2: Comparative study of the effectiveness of regularization in NADO training.

Compared to InsNet, it is much easier for an autoregressive model with NADO to handle flexible reordering/transformation of lexical constraints. This is reflected in the performance comparison of InsNet and NADO on CommonGen dataset. Under most settings, a Seq2seq base model makes it easier for the framework to perform well, as it guarantees a reasonable level of lexical constraint coverage in even the initial state of the model.

Using a DA pretrained base model is a even challenging setup since the lexical constraints are only imposed with NADO. Therefore, the base model distribution is much distinct from the one filtered by the oracle, which is shown by poor performances on both metrics. However, with warmup and NADO under importance sampling, we show that it is still possible to obtain a powerful model with the proposed method.

To further study the correlation between the base model quality and the improvement of NADO, we conduct experiments on GPT-2 base model. The GPT-2 base model has lower scores with and without NADO compared with GPT-2 large, while the coverage improvements are similar. It shows NADO is capable to push the base model distribution towards the oracle if the base model has decent quality.

We also do human evaluation on base model (GPT-2 Large fine-tune) and the best NADO system, together with the gold reference for comparison. The results are shown in Tab. 2. The evaluation metrics are detailed described in the Appendix. Some qualitative are shown in Tab. 3.

To study the importance of the regularization term, we conduct an ablative study under the optimal setting on the CommonGen dataset (Seq2seq base model with NADO only). The results are shown in Figure 2. While the success of achieving lexical control does not degenerate when NADO w/o regularization overfits, adding regularization can significantly improve the robustness of NADO generation quality when training NADO for more epochs.

## 4.2  Machine Translation with Formality Change

**Datasets and Setup**  We follow the experimental setting in FUDGE (Yang and Klein, 2021) to formalize the results of machine translation. Given an informal source sentence, our goal is to translate it into formal sentence written in the target language. We conduct our experiments on Fisher and CALLHOME Spanish-English Speech Translation Corpus (Post et al., 2013), where both of the Spanish source and English reference are informal and casual. Instead of evaluating the translation on original references, we use the formal and fluent rewritten version of references (Salesky et al., 2019) to evaluate the translation quality by BLEU scores. In the training process, the formal version reference is unseen to the models. We also evaluate the formality scores by a discriminator trained on GYAFC formality dataset (Rao and Tetreault, 2018) as what FUDGE paper does. In this experiment, pre-trained Marian MT model (Junczys-Dowmunt et al., 2018) is used as the base model.

In FUDGE, the authors train an auxiliary model also on GYAFC modeling token-level guidance $P(\text{formal}|\mathbf{y}_{<i})$, and leverage it to guide the base model by Bayesian rule

$$P(y_i|\mathbf{y}_{<i}, \text{formal}) \propto P(y_i|\mathbf{y}_{<i})P(\text{formal}|\mathbf{y}_{\leq i}). \qquad (9)$$

For the formality supervision, FUDGE leverages an external token-level oracle. In NADO, we load the same oracle but exclusively leverage sequence-level binary supervision as oracle $C$. We randomly

Table 1: Unsupervised/Supervised Lexically Constrained Generation results on Yelp Review (unsupervised), News (unsupervised) and CommonGen (supervised) dataset. CVRG stands for constraints coverage. For insertion-based models, on CommonGen dataset we directly use the keyword as initial context with no further permutation. $p, q$ denote the base model and the combined model in our framework, respectively. The domain adaptation pretrained model produces samples unconditioned on the constraints, and thus results in worse results than other setups. Results with * mark are from the open leader board on the test set instead of development set.

| Dataset | Yelp Review (test) | | News (test) | | CommonGen (dev) | |
|---|---|---|---|---|---|---|
| Metrics | BLEU-2/4 | CVRG | BLEU-2/4 | CVRG | BLEU-3/4 | CVRG |
| **Insertion-based Baselines** | | | | | | |
| InsNet-Sequential (Lu et al., 2022a) | 19.4/5.8 | 100% | 16.3/5.0 | 100% | 26.2/18.7 | 100% |
| ConstLevT (Susanto et al., 2020) | 14.8/4.0 | 100% | 11.8/1.9 | 100% | 21.3/12.3* | 96.9%* |
| **Algorithmic Baselines** | | | | | | |
| GPT-2-Large Finetune + Sampling | 16.4/5.3 | 94.5% | 13.2/4.2 | 81.8% | 34.2/24.7* | 82.2%* |
| Neural Logic (Lu et al., 2022a) | - | - | - | - | 36.7/26.7* | 97.7%* |
| A*esque Decoding (Lu et al., 2022b) | - | - | - | - | -/28.2* | 97.6%* |
| **Model Setups (Ours)** | | | | | | |
| $p$ (Domain Adaptation pretrain) | 5.3/0.4 | 5.4% | 4.0/0.8 | 0.9% | 9.3/3.9 | 8.5% |
| $p$ (Seq2seq pretrain) | 16.6/4.8 | 91.2% | 13.0/3.4 | 74.0% | 34.2/23.5 | 87.0% |
| $q$ (DA pretrained $p$ + warmup) | 16.2/4.3 | 75.4% | 12.6/2.8 | 66.7% | 32.7/20.9 | 79.7% |
| $q$ (DA pretrained $p$ + warmup + NADO) | 16.9/5.4 | 95.6% | **15.4/4.7** | **92.3%** | 37.8/26.2 | 96.1% |
| $q$ (Seq2seq pretrained $p$ + warmup) | 16.8/5.7 | 94.2% | 13.6/4.2 | 85.0% | 35.2/24.8 | 90.2% |
| $q$ (Seq2seq pretrained $p$ + NADO) | **17.4/6.0** | **96.7%** | 15.0/4.5 | 91.9% | **40.9/30.8** | **97.1%** |
| $q$ (Seq2seq pretrained $p$ + warmup + NADO) | 16.7/4.7 | 92.8% | 14.4/4.4 | 86.1% | 40.2/30.3 | 95.9% |
| **GPT-2 Base Reference** | | | | | | |
| $q$ (Seq2seq pretrained $p$) | - | - | - | - | 32.17/22.98 | 76.8% |
| $q$ (Seq2seq pretrained $p$ + NADO) | - | - | - | - | 33.61/24.01 | 85.5% |

Table 2: Human evaluation of generated texts in CommonGen test set. The detailed description for the four metrics (scale: from 1 to 3) and the evaluation setups can be found in the Appendix. Baseline stnads for GPT-2 Large fine-tune setting, and NADO stands for the best system, Seq2seq pretrained + NADO. We also evaluate the first gold reference provided in the dataset for comparison. NADO outperforms base model in all four metrics. (The difference is statistical significant tested by Wilcoxon signed ranks one-sided test, $p$-value $< 0.02$)

| Model | Quality | Plausibility | Concepts | Overall |
|---|---|---|---|---|
| Baseline | 2.39 | 2.46 | 2.40 | 2.37 |
| NADO (Ours) | 2.51 | 2.52 | 2.52 | 2.47 |
| Gold Ref. | 2.53 | 2.58 | 2.59 | 2.56 |

choose 10,000 (7.2%) source texts from the training set as input examples, and sample 8 target texts by sampling with temperature $T$ from base model $p$ for each source text. We use those sampled examples to train NADO. In total, we have $80,000$ training samples, which is similar to the number of training data (105k) for the token-level oracle in FUDGE. All the methods are using greedy decoding.

**Results and Discussion** The experimental results are shown in Table 4. Compared to FUDGE, although only the sequence-level supervision is leveraged, we are consistently better in both metrics, especially in BLEU score we boost about 3 points. We conjecture that the improvement is because our formulation is more principle and correct. In methods using auxiliary model to guide the base model, including FUDGE, their formulation is based on Eq. 9. However, the auxiliary model is trained on a distribution different from where the base model is pretrained on, which leads to a distributional discrepancy issue. In other words, directly multiplying these two terms is not rigorous, since they are estimated on two different distributions. On the contrary, NADO is trained specifically to the base model. This avoids the discrepancy issue and provides an accurate guidance. Considering we are

Table 3: Some more qualitative generation results with randomly selected concepts about NeurIPS.

| Constraint: | The generated texts should contain all the given concepts in arbitrary order |
|---|---|
| Concepts | **look forward discuss NeurIPS** |
| Base Model Sample #1 | Players **discuss** the **look** of **forward** NeurrIPS. (**NeurIPS**) |
| Base Model Sample #2 | Football player and **forward discuss** a **look** at the move. (**NeurIPS**) |
| NADO Sample #1 | People **look forward** to **discussing** the future of **NeurIPS**. |
| NADO Sample #2 | We **look forward** to meeting and **discussing** the future of **NeurIPS**. |
| Concepts | **excite paper accept NeurIPS** |
| Base Model Sample #1 | Researchers are **excited** after **acceptance** of their **paper** at IPS. (**NeurIPS**) |
| Base Model Sample #2 | Scientists **excited** to **accept paper accepted** at **NeurIPS**. |
| NADO Sample #1 | **NeurIPS** is **excited** to **accept** the **paper** of researcher. |
| NADO Sample #2 | **NeurIPS** is **excited** to announce that it has **accepted papers**. |

Table 4: Formal Machine Translation results. We follow (Yang and Klein, 2021) setting to choose BLEU score and average formality scores as the metric. We slightly improve the formality score compared to FUDGE, while significantly boost the BLEU score.

| Method | BLEU | Avg. Formality |
|---|---|---|
| MarianMT (Junczys-Dowmunt et al., 2018) | 16.98 | 0.45 |
| FUDGE (Yang and Klein, 2021) | 17.96 | 0.51 |
| NADO + Random Sampling | 20.84 | 0.54 |
| NADO + Sampling with $T = 5/4$ | 21.04 | 0.53 |
| NADO + Sampling with $T = 5/3$ | 20.77 | 0.52 |

using the same oracle function and similar number of training samples, the higher generation quality reflected by BLEU scores supports our conjecture.

In sampling, for each input we sample 8 examples to train $R_\theta^C$, which are usually identical in this task. Applying temperature in sampling allows NADO to be trained with more diverse data. Results show that with a properly set temperature, we can further improve the generation quality.

It is still possible that the neural oracle leverages some superficial or even spurious features and NADO is catering those features in order to improve the formality scores. For example, some informal little words like "hmm" "uh", and some abbreviations like " 'cause " "gonna" could make the formality score lower. We find that NADO tends to fix them (see Appendix D). However, how to get an good oracle is orthogonal to our contributions.

# 5   Conclusion

We purpose a general and efficient framework for controllable generation. We leverage an auxiliary neural model, NADO, to approximate the decomposed oracle guidance, and incorporate it with a fixed base model. By training with sampled data from the base model, NADO aligns better with the base model, and our framework is more flexible dealing with various application scenarios provided by different sampling methods. As NADO is a general framework, in the future, we plan to apply it in boarder application scenarios. For example, reducing societal bias (Sheng et al., 2019) (e.g., gender or racial bias) in generation by providing corresponding oracle.

# Acknowledgement

We thank anonymous reviewers for their comments and suggestions. We also thank members at UCLANLP and UCLAPLUS labs for their feedback. The project is supported in part by CISCO, Amazon, Sloan foundation, Google, and DARPA MCS program under Cooperative Agreement N66001-19-2-4032. In addition, Tao is supported as an Amazon Fellow.

## Limitation

In this work we assume that a base model with decent quality (e.g., large pretrained language models) and a good oracle for controlling attributes are available. However, in some applications, the quality of the base model may be low and the oracle may only capture superficial shortcut between constraints and labels. How to control a generation model under these situations is an interesting future work direction.

Similar to other language generation approaches, we note that there is a risk that malicious users may use NADO to generate improper or toxic texts. Also, the generated text may contain societal biases inherited from data. However, on the other hand, NADO provides a powerful weapon against toxicity as developers can design constraints to detoxify the generated text. We refer readers to the discussion in Sheng et al. (2019, 2021); Zellers et al. (2019); Bender et al. (2021); Radford et al. (2019); Brown et al. (2020); Dev et al. (2021); Dhamala et al. (2021).

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
