# A  Closed-form Token-level Decomposition

## A.1  Decomposition with Hard Constraints

The sequence-level solution $q^*$ is given by

$$q^*(\mathbf{y}|\mathbf{x}) = \frac{p(\mathbf{y}|\mathbf{x})C(\mathbf{x},\mathbf{y})}{R_p^C(\mathbf{x})}.$$

Now we prove that

$$q^*(y_i|\mathbf{x},\mathbf{y}_{<i}) = \frac{R_p^C(\mathbf{x},\mathbf{y}_{\leq i})}{R_p^C(\mathbf{x},\mathbf{y}_{\leq i-1})}p(y_i|\mathbf{x},\mathbf{y}_{<i}),$$

is the unique token-level decomposition. On one hand, we verify $q^*$ is a valid decomposition, which can be demonstrated by

$$
\begin{aligned}
\prod_{i=0}^{L} q^*(y_i|\mathbf{x},\mathbf{y}_{<i}) &= \prod_{i=1}^{L} \frac{R_p^C(\mathbf{x},\mathbf{y}_{\leq i})}{R_p^C(\mathbf{x},\mathbf{y}_{\leq i-1})}p(y_i|\mathbf{x},\mathbf{y}_{<i}) \\
&= \frac{R_p^C(\mathbf{x},\mathbf{y}_{\leq L})}{R_p^C(\mathbf{x},\mathbf{y}_{\leq 0})} \prod_{i=0}^{L} p(y_i|\mathbf{x},\mathbf{y}_{<i}) \\
&= \frac{C(\mathbf{x},\mathbf{y})}{R_p^C(\mathbf{x})}p(\mathbf{y}|\mathbf{x}) \\
&= q^*(\mathbf{y}|\mathbf{x}),
\end{aligned}
\tag{10}
$$

together with

$$\sum_{y_i} q^*(y_i|\mathbf{x},\mathbf{y}_{<i}) = \frac{\sum_{y_i} R_p^C(\mathbf{x},\mathbf{y}_{\leq i})p(y_i|\mathbf{x},\mathbf{y}_{<i})}{R_p^C(\mathbf{x},\mathbf{y}_{\leq i-1})} = 1 \tag{11}$$

On the other hand, we demonstrate that the decomposition is unique. We generally prove that

**Lemma 3** For finite space $\mathcal{X} \times \mathcal{Y}$, if the sequence-level distribution $q(\mathbf{y}|\mathbf{x})$ is determined, the token-level distribution is unique.

**Proof** We assume the longest sequence in $\mathcal{Y}$ has length $L$, and pad all the sequence to length $L$ by adding special token at the end. We prove the following statement by induction:

Given input $\mathbf{x}$ and prefix length $i$, If $q(y_i|\mathbf{x},\mathbf{y}_{<i})$ is determined, there exists a unique distribution $q(y_{i-1}|\mathbf{x},\mathbf{y}_{<(i-1)})$.

When $i = L$, it is true since $y_i \oplus \mathbf{y}_{<i}$ is the full sequence ($\oplus$ denotes the string concatenation), hence $q(y_i|\mathbf{x},\mathbf{y}_{<i}) \propto q(y_i \oplus \mathbf{y}_{<t}|\mathbf{x})$. $q(y_i \oplus \mathbf{y}_{<t}|\mathbf{x})$ is the determined sequence-level distribution, so $q(y_i|\mathbf{x},\mathbf{y}_{<i})$ is unique.

Assume the statement holds when $i >= t$, we consider $i = t - 1$. The distribution of prefixes with length $t - 1$ is given by

$$q(\mathbf{y}_{<(t-1)}|\mathbf{x}) \propto \frac{q(\mathbf{y}|\mathbf{x})}{\prod_{j=t}^{L} q(y_j|\mathbf{x},\mathbf{y}_{<j})}.$$

Since the sequence distribution and token-level distribution after step $t$ are determined, the prefix distribution is also determined. Thus, the token-level distribution at step $(t - 1)$ has unique solution

$$q(y_{t-1}|\mathbf{x},\mathbf{y}_{<(t-1)}) \propto q(\mathbf{y}_{<(t-1)}|\mathbf{x}).$$

## A.2  Closed-form Solution and Decomposition with Soft Constraints

To deal with soft constraints, we define the feasible set $Q$ as

$$Q := \{q|\sum_{\mathbf{y}:\, C(\mathbf{x},\mathbf{y})=0} q(\mathbf{y}|\mathbf{x}) = r\}.$$

The sequence-level solution is given b

$$q^*(\mathbf{y}|\mathbf{x}) = \arg\min_{q \in Q} KL(p(\mathbf{y}|\mathbf{x}) \| q(\mathbf{y}|\mathbf{x}))$$

$$= \arg\min_{q \in Q} \left[ \sum_{\mathbf{y}:C(\mathbf{x},\mathbf{y})=0} p(\mathbf{y}|\mathbf{x}) \log \frac{q(\mathbf{y}|\mathbf{x})}{p(\mathbf{y}|\mathbf{x})} + \sum_{\mathbf{y}:C(\mathbf{x},\mathbf{y})=1} p(\mathbf{y}|\mathbf{x}) \log \frac{q(\mathbf{y}|\mathbf{x})}{p(\mathbf{y}|\mathbf{x})} \right]$$

The optimal distribution $q^*$ should be proportional to $p$ in both term, respectively. Since $\sum_{\mathbf{y}:C(\mathbf{x},\mathbf{y})=1} p(\mathbf{y}|\mathbf{x}) = R_p^C(\mathbf{x})$, we have

$$q^*(\mathbf{y}|\mathbf{x}) = \begin{cases} \frac{1-r}{1-R_p^C(\mathbf{x})} p(\mathbf{y}|\mathbf{x}) & C(\mathbf{x},\mathbf{y}) = 0 \\ \frac{r}{R_p^C(\mathbf{x})} p(\mathbf{y}|\mathbf{x}) & C(\mathbf{x},\mathbf{y}) = 1 \end{cases}$$

We denote $\alpha = \frac{r}{R_p^C(\mathbf{x})}$, $\beta = \frac{1-r}{1-R_p^C(\mathbf{x})} p(\mathbf{y}|\mathbf{x})$. Now the sequence-level optimal distribution is determined, by Lemma 3, we have a unique token-level decomposition. Now we verify

$$q^*(y_i|\mathbf{x},\mathbf{y}_{<i}) = \frac{\alpha R_p^C(\mathbf{x},\mathbf{y}_{\leq i}) + \beta(1 - R_p^C(\mathbf{x},\mathbf{y}_{\leq i}))}{\alpha R_p^C(\mathbf{x},\mathbf{y}_{\leq i-1}) + \beta(1 - R_p^C(\mathbf{x},\mathbf{y}_{\leq i-1}))} p(y_i|\mathbf{x},\mathbf{y}_{<i})$$

is exactly what we want. We have

$$\prod_{i=1}^{L} q^*(y_i|\mathbf{x},\mathbf{y}_{<i}) = \prod_{i=1}^{L} \frac{\alpha R_p^C(\mathbf{x},\mathbf{y}_{\leq i}) + \beta(1 - R_p^C(\mathbf{x},\mathbf{y}_{\leq i}))}{\alpha R_p^C(\mathbf{x},\mathbf{y}_{\leq i-1}) + \beta(1 - R_p^C(\mathbf{x},\mathbf{y}_{\leq i-1}))} p(y_i|\mathbf{x},\mathbf{y}_{<i})$$

$$= \frac{\alpha R_p^C(\mathbf{x},\mathbf{y}_{\leq L}) + \beta(1 - R_p^C(\mathbf{x},\mathbf{y}_{\leq L}))}{\alpha R_p^C(\mathbf{x}) + \beta(1 - R_p^C(\mathbf{x}))} \prod_{i=1}^{L} p(y_i|\mathbf{x},\mathbf{y}_{<i})$$

$$= \frac{\beta + (\alpha - \beta) C(\mathbf{x},\mathbf{y})}{r + (1 - r)} p(\mathbf{y}|\mathbf{x})$$

$$= q^*(\mathbf{y}|\mathbf{x}),$$

and

$$\sum_{y_i} q^*(y_i|\mathbf{x},\mathbf{y}_{<i}) = 1,$$

since

$$\sum_{y_i} R_p^C(\mathbf{x},\mathbf{y}_{\leq i})) p(y_i|\mathbf{x},\mathbf{y}_{<i}) = R_p^C(\mathbf{x},\mathbf{y}_{\leq i-1}).$$

Therefore, $q^*$ is the unique solution.

## B    Proof of the Error Bounds

Here we prove the bounds in Lemma 1 and Lemma 2. [6]

**Proof of Lemma 1**    We denote $Z_i(\mathbf{x},\mathbf{y})$ as the normalization term in distribution $q(y_i|\mathbf{x},\mathbf{y}_{<i})$, i.e.,

$$q(y_i|\mathbf{x},\mathbf{y}_{<i}) = \frac{1}{Z_i(\mathbf{x},\mathbf{y})} \frac{R_\theta^C(\mathbf{x},\mathbf{y}_{\leq i})}{R_\theta^C(\mathbf{x},\mathbf{y}_{\leq i-1})} p(y_i|\mathbf{x},\mathbf{y}_{<i}),$$

$$Z_i(\mathbf{x},\mathbf{y}) = \sum_{y_i} \frac{R_\theta^C(\mathbf{x},\mathbf{y}_{\leq i})}{R_\theta^C(\mathbf{x},\mathbf{y}_{\leq i-1})} p(y_i|\mathbf{x},\mathbf{y}_{<i}).$$

Since $\frac{1}{\delta} < \frac{R_\theta^C(\mathbf{x},\mathbf{y}_{\leq i})}{R_p^C(\mathbf{x},\mathbf{y}_{\leq i})} < \delta$,

$$Z_i(\mathbf{x},\mathbf{y}) < \sum_{y_i} \delta^2 \frac{R_p^C(\mathbf{x},\mathbf{y}_{\leq i})}{R_p^C(\mathbf{x},\mathbf{y}_{\leq i-1})} p(y_i|\mathbf{x},\mathbf{y}_{<i}) = \delta^2.$$

---

[6]There are typos in the main text. Here we correct them. The typos do not affect related conclusions.

Similarly, $Z_i(\mathbf{x}, \mathbf{y}) > \frac{1}{\delta^2}$.

The sequence-level distribution is given by

$$q(\mathbf{y}|\mathbf{x}) = \prod_{i=1}^{L} q(y_i|\mathbf{x}, \mathbf{y}_{<i}) = \frac{1}{\prod_{i=1}^{L} Z_i(\mathbf{x}, \mathbf{y})} \frac{R_\theta^C(\mathbf{x}, \mathbf{y}_{\leq L})}{R_\theta^C(\mathbf{x})} p(\mathbf{y}|\mathbf{x}).$$

Now we bound the KL-divergence as

$$
\begin{aligned}
KL(q^*(\mathbf{y}|\mathbf{x}) \| q(\mathbf{y}|\mathbf{x})) &= \sum_{\mathbf{y} \in \mathcal{Y}} q^*(\mathbf{y}|\mathbf{x}) \log \left( \frac{q(\mathbf{y}|\mathbf{x})}{q^*(\mathbf{y}|\mathbf{x})} \right) \\
&= \sum_{\mathbf{y}: C(\mathbf{x}, \mathbf{y})=1} q^*(\mathbf{y}|\mathbf{x}) \log \left( \frac{1}{\prod_{i=1}^{L} Z_i(\mathbf{x}, \mathbf{y})} \frac{R_\theta^C(\mathbf{x})}{R_p^C(\mathbf{x})} \frac{C(\mathbf{x}, \mathbf{y})}{R_\theta^C(\mathbf{x}, \mathbf{y}_{\leq L})} \right) \\
&< \sum_{\mathbf{y}: C(\mathbf{x}, \mathbf{y})=1} q^*(\mathbf{y}|\mathbf{x}) \log(\delta^{(2L+2)}) \\
&< (2L+2) \log \delta.
\end{aligned}
$$

**Proof of Lemma 2** With additional condition, $Z_i(\mathbf{x}, \mathbf{y}) = 1$.. Hence the bound is $2 \log \delta$.

## C  Experiment Details

### C.1  Datasets

In MT formality change experiment, we use Fisher and Callhome Spanish-English translation dataset (Post et al., 2013) and GYAFC formality corpus (Rao and Tetreault, 2018). Both of the datasets are not public. Fisher and Callhome dataset is owned by LDC. To acquire the GYAFC dataset, one need to first gain access to Yahoo Answers corpus.

For unsupervised LCG experiments, we use Yelp Reviews (Cho et al., 2018) and WMT News section datasets (Bojar et al., 2017; Guo et al., 2018). Yelp Reviews is published under its own license [7]. Please refer to the official website of WMT dataset (Bojar et al., 2017) for more information about licenses and legal concerns. We conduct our experiment of supervised LCG on the CommonGen dataset (Lin et al., 2020), whose official repository is published under MIT license.

### C.2  Hyperparameters and Training Details

For MT experiments, we load the MarianMT from the es-en checkpoint provided by huggingface. We set a constant learning rate $1e-5$. The NADO shares the encoder with MarianMT, and use MarianDecoder as the decoding architecture. Compared to the 12-layer decoder in MarianMT, we a use 3-layer decoder, and the other configurations follow the Marian decoder. The fully connected output layer is zero initialized. We set the coefficient $\lambda = 0.1$ for the regularization, which is selected from $\{0.01, 0.03, 0.1, 0.3, 1.0\}$. All the hyperparameters are tuned on the development set. We select the best NADO model evaluated on development set in 10-epoch training.

For LCG experiments, we finetune our base models from a GPT-2-Large checkpoint (Radford et al., 2019) for 3 epochs with learning rate $1e-5$. We use a warmup number of $400$ and the learning rate decays to 0 in 5000000 steps (which are far more than the actually executed ones). We use NADO to train the auxiliary $R$ models (4-layer, 768-D, 12 headed transformer) with a learning rate of $2e-5$, which is selected from $\{1e-5, 2e-5, 5e-5\}$. We zero-initialize the output layer of auxiliary models. No learning warmup/decay is applied when training $R$. With the help of KL-regularization, we set $\lambda = 1.0$ and we don't observe very severe overfitting in the second stage of NADO training. We simply report the results after the maximum number of training epochs (usually 20).

For more implementation details and tricks, please refer to our code. [8]

---

[7]See full license file here.

[8]https://github.com/MtSomeThree/constrDecoding

Table 5: Some qualitative generation results in machine translation formality change experiments. NADO controls the model to be more formal by skipping informal words, fixing the grammar, capitalizing the first letters in base model generation results, etc. Noting that the zero-shot base model can not handle the repeated words in casual source language very well, we observe that the NADO-controlled model, which follows the base model distribution, cannot handle it, either.

| Constraint: | The generated (translated) texts should be fluent and formal. |
|---|---|
| Source | Lo, y las eh usted dice que usted toca instrumentos, eh es- ¿usted su esposo tocan como en una banda, un grupo? |
| MarianMT | **What**, and **the uh, you say** you play instruments, uh, you're– you're your husband playing like a band, a band? |
| NADO | **So**, and **you're saying that** you play instruments, uh, you're– you're your husband playing like **in** a band, a band? |
| Formal Ref. | So, you say you play instruments, you and your husband play in a band or group? |
| Source | Entonces, ya los programas ya, ya están clasificados, ya ver una escena de de pornografía en la televisión a cualquier hora del día es normal. Y ya para muchos |
| MarianMT | So, **already the programs are already classified**, and seeing a porn scene on TV at any time of the day is normal. **And already for many** |
| NADO | So, **the programs already, they're already classified**, and seeing a porn scene on TV at any time of the day is normal. |
| Formal Ref. | The programs, are classified, to see a scene of pornography on the television at anytime of the day is normal. And for many |
| Source | donde hay problemas |
| MarianMT | **where** there's trouble. |
| NADO | **Where** there's trouble. |
| Formal Ref. | That's when trouble arises. |

## C.3 Computational Resources

We run the MT experiments on a NVIDIA GeForce RTX 2080 Ti GPU. We run LCG experiments on a NVIDIA A6000 GPU.

# D Qualitative Generation Results

## D.1 Machine Translation Formality Change Experiments

In Tab.5 we show some translation text in the best system (NADO+Sampling with $T = 5/4$) and baseline (MarianMT), together with the source and its formal reference.

## D.2 Lexical Constrained Generation Experiments

In Tab.5 we show some generated text in best systems and baselines, together with the golden references.

# E The Human Evaluation Setups

We use the template shown in Fig. 3 to do the human evaluation. We sampled 993 keys from CommonGen test set and generate sentences. For each sentence we ask two MTurkers to annotate. We filter the MTurkers by approval rate greater than 97% and the number of approval greater than 50. The pay rate is $0.05 per sentence.

**Read the given concepts and sentence below and use the sliders below indicate how much you agree with the statements. (1=Yes, 2=Somewhat, 3=No)**

Concepts: Hit2_concepts_data

Sentence: Hit2_sentence_data

- 1) **Sentence Quality**: Is the **sentence** *well-formed*.

  **Yes**: The sentence is **well-formed** and **fluent**.

  **Somewhat**: The sentence is **understandable** but a bit awkward.

  **No**: The sentence is **neither** well-formed or fluent.

  ○———————————

  (1=**Yes**, 2=**Somewhat**, 3=**No**)

- 2) **Plausibility**: Does the **sentence** describe a plausible scenario.

  **Yes**: The sentence describes a **realistic** or **plausible** scenario.

  **Somewhat**: The sentence describes an **acceptable** scenario but a bit awkward.

  **No**: The sentence describes a **nonsensical** scenario.

  ○———————————

  (1=**Yes**, 2=**Somewhat**, 3=**No**)

- 3) **Concepts**: Does the **sentence** include the given **concepts** meaningfully.

  **Example:** if "run" is a given concept, sentence should include word "ran", "running" or other variant forms of "run". Synonyms like "jog" are **not** allowed.

  **Yes**: The sentence **meaningfully** includes **all** of the concepts.

  **Somewhat**: The sentence meaningfully includes some, but not all of the concepts. Or, the sentence includes all concepts but some of them are not meaningful or prope

  **No**: The sentence **does not** include concepts in a meaningful way.

  ○———————————

  (1=**Yes**, 2=**Somewhat**, 3=**No**)

- 4) **Overall**: Considering your answers to 1), 2) and 3), does the **sentence** meaningfully combine all of the **concepts** into a well-formed and plausible scenario?.

  **Yes**: The sentence is reasonably well-formed/understandable, and meaningfully combines **all** the concepts into a plausible scenario.

  **Somewhat**: The sentence looks okay in terms of above questions

  **No**: The sentence is not well-formed/understandable, or fails to properly combine **all** the concepts into a plausible scenario.

  ○———————————

  (1=**Yes**, 2=**Somewhat**, 3=**No**)

Submit

Figure 3: Human evaluation template for LCG experiments. To make the evaluation results easy to read, we score "Yes" for 3 points, "Somewhat" for 2 points and "No" for 1 point.

Table 6: Some qualitative generation results in LCG benchmark experiments. NADO focuses on making sure all the constraints are imposed, or at least ensuring as many as possible of them. This is in particular more obvious when using weaker base models (e.g. GPT-2-base).

| Constraint: | The generated texts should contain all the given concepts in arbitrary order |
| --- | --- |
| Concepts | **kid room dance** |
| Base Model | A boy and girl **dancing** in a **room**. (~~**kid**~~) |
| NADO | A **kid** is **dancing** in the **room**. |
| Golden Ref. #1 | The silly **kid** loves to **dance** in her **room**. |
| Golden Ref. #2 | the **dance kid**: **room** is full of **kids** |
| Golden Ref. #3 | A **kid** is **dancing** in the **room**. |
| Golden Ref. #4 | A group of **kids** are **dancing** around a living **room**. |
| Concepts | **create pottery wheel** |
| Base Model | add a **pottery wheel** to your home. (~~**create**~~) |
| NADO | **create** a **pottery wheel** in the garden |
| Golden Ref. #1 | **Create pottery** with a **wheel** and clay. |
| Golden Ref. #2 | I **create pottery** on a **wheel**. |
| Golden Ref. #3 | A **pottery wheel** can be used to **create** bowls. |
| Golden Ref. #4 | A man is using a **wheel** to **create pottery**. |
| Concepts | **bed look sit** |
| Base Model | A man is **sitting** on a **bed**. (~~**look**~~) |
| NADO | A man **sits** on a **bed** and **looks** at his reflection in the mirror. |
| Golden Ref. #1 | The girl was **sitting** in the couch and **looking** at a bed in a catalog. |
| Golden Ref. #2 | the woman **look** at her **bed** as she **sits**. |
| Golden Ref. #3 | The man **sat** on the **bed** and **looked** out the window |
| Golden Ref. #4 | **Looking** dejected, someone **sits** on his **bed**. |