# OpenReview forum: "Controllable Text Generation with Neurally-Decomposed Oracle"
_NeurIPS.cc/2022/Conference — NeurIPS 2022 Accept_

### Official Review · Reviewer_B6Lr · 2022-07-09

**Rating:** 7
**Confidence:** 3
**Soundness:** 4 excellent
**Presentation:** 3 good
**Contribution:** 3 good

**Summary:**

This paper proposes to decompose sequence-level attributes into token-level guidance for controllable text generation, where the token-level guidance is approximate by a neural model trained with examples sampled from the base model. Both theoretical analysis and experimental results demonstrate the effectiveness of the proposed method on different tasks.

**Questions:**

Refer to comments above.

**Limitations:**

Yes. I do not see any limitations.

**Strengths And Weaknesses:**

Pros:
- This paper is well-written and easy to read.
- The proposed method is novel and well supported by theoretical analysis
- Experiments are exhaustive. The authors conduct experiments on both constrained text generation and machine translation, and show the performance improvement on various datasets.

Cons:
- The proposed NADO model introduces some additional hyperparameters that need to be tuned. It’s better if the authors can provide a deep analysis on the effect/sensitiveness of hyperparameters.

---

> ### Author Response · Authors · 2022-08-02
> **Response to Reviewer B6Lr**
>
> Thanks for your comments.
>
> _"Hyperparameters in NADO”:_  Our key innovation is on how to incorporate constraints in the decoding, and for that stage, there is no hyper-parameter needed as we derive an optimal closed-form solution based on posterior regularization. This is in contrast to neural logic decoding, which requires some hyper-parameters to control the strength of constraints, and FUDGE/GeDI, which requires hyper-parameters to balance the logits between the base and the auxiliary model.
>
> However, there are hyper-parameters involved in 1) sampling strategy and 2) model design for R_\theta. For the former, we show that the results are not sensitive to the temperature in Sec 3.5, and we are able to select it by evaluation on the development set. For the latter, we follow the standard design of neural models to select the model architecture, number of layers, etc. The same process is required in earlier works leveraging auxiliary models like FUDGE.

---

### Official Review · Reviewer_2y5z · 2022-07-11

**Rating:** 7
**Confidence:** 4
**Soundness:** 3 good
**Presentation:** 3 good
**Contribution:** 3 good

**Summary:**

This work proposes controllable autoregressive generation by decomposition the control signal into token level constraints. The method is formulated as an optimization problem based on posterior regularization and approximated by a neural network. Experiments on lexically controlled generation and machine translation with formality demonstrates the effectiveness of the method.

**Questions:**

See above comments.

**Limitations:**

How the method relate to or can help improper language generation? It would be nice if the authors add certain discussions about how improper sentences can be prevented with this method.

**Strengths And Weaknesses:**

Strengths

- This is indeed a novel method for controlling autoregressive generation. The method is different than the existing paradigms like lexically controlled beam search or modified sampling probability. The authors also provide a certain level of theoretical guarantee.
- This method is demonstrated effective in the experiments, especially the machine translation with formality control.

Weakness

- There are important missing details, specifically:
    - How many samples are need for the approximation (Section 3.3)? Intuitively, as the approximation is performed over the full autoregressive decoding space, one would expect the number of sample be large, thus the computation complexity may also be large.
    - How good and diverse should the base model be? If the base model is not good enough for generation, then the controlled q will consequently not be good. If the base model is not diverse enough, then one may require more sampling to hit the condition (note that controlling the temperature cannot alleviate the mode collapse problem, if it is the problem of the base model)
    - How does the performance related to model scale? Can one expect a larger model to be easier to control (possibly because of better language modeling) or harder to control (possibly because of harder optimization)?
    - What if base distribution is far from constraints (but may sill be good per se) such that the sampled sentences cannot hit the constraints? This could happen in settings where the constraint distribution is far from the model distribution.
    - What happens if the sampled sequence partially meet the constraint? How would the method behave and can it still learn from partial satisfaction?
- My other concern is that the experiments not grounded to linguistic explanations. Specifically:
    - How does the generated sequences look like? I would encourage the authors to include examples about the intermediate sample of q and show how q start from not following the constraints and gradually becomes more following.
    - How does the model differ from other model in terms of generated sequence?
- My final concern is that there is no human evaluation. Generally, classifier-based evaluation (Table 2) may be fooled by certain spurious correlations thus not strongly reliable. Can the authors include human evaluation, or at least put generated examples for the reviewers to get an impression?

I will be happy to increase my score accordingly if the above concerns can be properly addressed.

---

> ### Author Response · Authors · 2022-08-02
> **Response to Reviewer 2y5z**
>
> Thanks for the comments and questions. Please see the response below:
>
> _"Number of samples needed for the approximation ”:_ The number of samples required to approximate the search space depends on the complexity of the constraints. However, in our experiments, we found that a reasonably large number of samples is sufficient to train a good NADO to capture the **inherent correlation between generated token and the sequence-level oracle**. In both experiments, we show that with the same small number of samples (80,000 samples for MT, and 35141 * 16 samples for CommenGen) as the auxiliary data, our approach is more effective than existing approaches such as FUDGE (see Sec. 4.2). We also discuss how to generate effective samples by important sampling in Sec 3.5.
>
> _“Requirements for base model quality / diverse”:_ Indeed, if the base models perform suboptimal, the quality of control will be influenced. This will be an issue for all other approaches. In practice, we consider the setting where the base model is a large pre-trained language model and has a decent quality as shown in the experiments. We also update some results using GPT2-base in CommenGen, please refer to the revision. Basically, the generation quality (evaluated by BLEU scores) drops a little bit compared to GPT2-large. Please refer to the response below to see the further discussion about controllability.
>
>
> _“What if base distribution is far from constraints”:_ This is a naturally challenging setting and it is exactly the scenario NADO contributes the most compared with the existing alternatives. Please refer to Table 1, the performance of p(Domain Adaptation pretrain) row. This is an example where the base model is far from constraints and hard to hit it (very low constraint coverage). With NADO we are able to boost the coverage to 96.1% leveraging importance sampling and warmup.
>
>
> _“Model scale vs. controllability”:_ Our preliminary results show that the controllability is not correlated with model scale. The controllability is much more sensitive to the complexity of oracle rather than the base model distribution. Imagine a simple constraint “never output the token ‘rejection’ “, it is easy to be learned by NADO no matter how large the base model is. What the base model scale/quality affects most is the generation quality after controlling. We include new results with GPT2-small in CommenGen, please refer to the revision. Basically, the GPT-2 base model has lower scores with and without NADO compared with GPT-2 large, while the coverage improvements are similar.
>
>
> _“Partially meet the condition”:_ It is possible to extend C as a real value function in [0,1] by reformulating our approach. Empirically, in LCG experiment, we conducted a preliminary experiment to define C as the keywords coverage and the results are a little bit worse than using a binary C: for example, in CommenGen experiment q (Seq2seq pretrained p + NADO) setting (the second row from last), if we change C from binary to continuous, we get coverage drop from 97.1 to 96.6, and BLEU-3/4 score drop from 40.9/30.8 to 40.1/29.9.
>
> _“Sequences look like, compared to other models”:_ We add some generation samples in the appendix D.
>
> _“Certain spurious correlations”_: Our analysis and derivation are based on the theoretical assumption that we have an oracle C; how to get such an oracle is orthogonal to our contributions. However, in practice, in LCG the oracle is a simple rule-based keywords checker and it is almost perfect. In MT the oracle is a neural network that may leverage some superficial features. For example, some informal little words like “hmm” “uh”, some abbreviations like “ ‘cause ” “gonna” and capital letters. We find that NADO tends to fix them (please refer to Appendix D). Generally it makes the sentences more fluent and formal so we believe the oracle is good.
>
> _“Human Evaluation”_: We agree that human evaluation can better tell the performance of language generation applications. However, our goal is to control the model generation with respect to the given oracle, while keeping the overall generation quality compared to the base model. Thus, we evaluated the approaches by automatic metrics (BLEU scores, oracle scores). This follows the setting of related work like neural logic decoding. We also provide some qualitative examples in the Appendix D.
>
> _“Improper language generation”_: Preventing improper language generation is a great application of our method. Our approach can be applied when having a high quality oracle that classifies the improper sentences (for example, PerspectiveAPI) or a blocklist of toxic phrases.

---

> ### Author Response · Authors · 2022-08-08
> **Specific Response to Reviewer 2y5z**
>
> Dear Reviewer 2y5z,
>
> Thanks for your valuable comments and we believe they help a lot in our revision. To address your concern, we provided further empirical results about a smaller base model (GPT2-base) and continuous oracle (partially satisfied), including related analysis and discussion. We clarified our assumptions and key challenges in application scenarios that we target. Some qualitative generation results were also provided in the appendix of the revision to help better understand the effect and generation quality of NADO. Basically, we can control the model generation with respect to the given constraints, while **keeping the overall generation quality comparable to the base model**.
>
> We hope our responses sufficiently addressed your concerns such that you will raise your rating as you mentioned in your original review, and we are more than happy to have a discussion if you have any follow-up questions or comments.

---

> > ### Comment · Reviewer_2y5z · 2022-08-09
> > **Thank you for your response**
> >
> > I thank the authors for their detailed response, revision, and additional results. The authors have addressed most of my questions, and I am happy to increase my score (also apologize for this last-minute response)

---

### Official Review · Reviewer_u95g · 2022-07-18

**Rating:** 7
**Confidence:** 3
**Soundness:** 4 excellent
**Presentation:** 3 good
**Contribution:** 4 excellent

**Summary:**

**What is the task?**
A general, flexible and efficient framework to control auto-regressive generation models with NeurAlly-Decomposed Oracle (NADO)


**What has been done before?**
PPLM , GeDI and FUDGE also aim to guide the base model with an auxiliary 42 model. However, they either shift the base model distribution in a post-hoc manner without theoretical guarantee, or/and require external labeled data to train the auxiliary model. Instead, we derive a closed-form solution for the optimal way to incorporate the oracle, without requiring external labeled data or token-level guidance.

**What are the main contributions of the paper?**

* Given a pre-trained base language model and a sequence-level boolean oracle function, author((s) propose to decompose the oracle function (indicating whether an attribute is satisfied) into token-level guidance to steer the base model in text generation.

* They present the closed-form optimal solution to incorporate the token-level guidance into the base model for controllable generation.

* They provide a theoretical analysis of how the approximation quality of NADO affects the controllable generation results.


**What are the main results?**
Experiments conducted on two applications: (1) text generation with lexical constraints and (2) machine translation with formality control demonstrate that our framework efficiently guides the base model towards the given oracle while maintaining high generation quality.



**Questions:**

Typo : Line 41 aims→ aim

**Limitations:**

There is no limitations section i the paper.

**Strengths And Weaknesses:**

Strengths

* Generally, the post-processing methods are considered expensive in inference and low quality in generated texts. However, proposed framework, as a kind of post-processing method, can achieve high generation quality demonstrated in experiments and is efficient in the inference time.

* Since NADO is trained on the data sampled from the base models, it aligns better with the base model and thus can achieve better control.

* The token-level guidance is approximated by a neural model trained with examples sampled from the base model, demanding no additional auxiliary labeled data.

---

> ### Author Response · Authors · 2022-08-02
> **Response to Reviewer u95g**
>
> Thanks for your comments. We have fixed the typos and added the limitation section in the revision as following.
>
> In this work, we assume a practical setting, where we have a base model with decent quality (e.g., large pretrained language models) and an oracle for the controlled attributes. We also assume we have the access to the probability distribution of generated tokens in each step by the base model, so that we can learn an auxiliary model to reweight the distribution. However, retraining base model is not required.
>
> We also note that similar to other language generation approaches, there is a risk that malicious users may use NADO to generate improper or toxic texts and the generated texts may contain societal biases inherited from data. However, on the other hand, NADO could be a powerful weapon against toxicity and biases by incorporating a blocklist of toxic phrases. We refer readers to the discussion in  Sheng et al. (2019); Zellers et al. (2019); Bender et al.(2021); Radford et al. (2019); Brown et al. (2020).

---

### Author Response · Authors · 2022-08-07
**Looking Forward to Further Discussion**

Dear Reviewers，

We appreciate your valuable comments and feedback. In our response we managed to address all of your concerns and provide related experimental results for some of them. Considering the end of author-reviewer discussion period is approaching (Aug. 9, next Tue), we would like to know if our responses are clear enough to resolve your concerns and we are open to further discussion.

---

### Meta-Review · Area_Chair_GyvG · 2022-08-25

**Recommendation:** Accept
**Confidence:** Certain

**Metareview:**

All three reviewers sided to accept the paper. The method of the paper is formulated as an optimization problem based on posterior regularization, and as such is quite different from existing paradigms in controllable NLG (e.g., lexically constrained beam search or modified probability sampling). The work's theoretical basis also offers a nice contrast with established methods in this area, as the existing methods are often applied in post-hoc manners and without theoretic guarantees. The only significant downside of this paper is that its evaluation is not very standard and lacks human evaluation, and its model-based automated evaluation of attributes such as formality could have been affected by spurious correlations (note: the latter concern affects only one of two tasks of the paper). As the paper achieves some substantial gains on two very different tasks, the reviewers generally considered the method of the paper to be quite effective.

**Award:**

No

---

### Decision · Program_Chairs · 2022-09-14

Accept